# Doctor for Every Citizen: Telehealth Visits at Dubai Health Authority during COVID-19 Pandemic in Dubai, United Arab Emirates

**DOI:** 10.3390/healthcare11030294

**Published:** 2023-01-18

**Authors:** Wafa K. Alnakhi, Heba Mamdouh, Hamid Y. Hussain, Mohamed S. Mudawi, Gamal M. Ibrahim, Amal J. Al Balushi, Noora Al Zarooni, Abdulsalam Elnaeem, Nabil Natafgi

**Affiliations:** 1Data Analysis, Research and Studies Department, Dubai Health Authority, Dubai 7272, United Arab Emirates; 2Mohammed bin Rashid University of Medicine and Health Sciences, Dubai 505055, United Arab Emirates; 3Department of Family Health, High Institute of Public Health, Alexandria University, Alexandria 21561, Egypt; 4Department of Data Analysis, High Institute for Management Sciences, Belqas 35631, Egypt; 5Maternal and Child Health Nursing Department, Oman College of Health Science, Muscat P.O. Box 407400, Oman; 6Department of Medicine, University of Central Lancashire, Preston PR1 2HE, UK; 7Department of Mathematics, Faculty of Science, University of Technology Malaysia, Johor Bahru 81310, Malaysia; 8Department of Health Services Policy & Management, Arnold School of Public Health, University of South Carolina, Columbia, SC 29201, USA

**Keywords:** telehealth, telemedicine, Dubai Health Authority (DHA), wait time, turnaround time, appointment completion

## Abstract

Background: Digital health significantly affects healthcare delivery. Moreover, empirical studies on the utilization of telehealth in Dubai are limited. Accordingly, this study examines the utilization of telehealth services in Dubai Health Authority (DHA) facilities and the factors associated with telehealth appointment completion and turnaround time. Methods: This cross-sectional study examines patients who used telehealth services in DHA from 2020 through 2021 using 241,822 records. A binary logistic regression model was constructed to investigate the association between appointment turnaround time as a dependent variable and patient and visit characteristics as independent variables. Results: Of the total scheduled telehealth visits, more than three-quarter (78.55%) were completed. Older patients, non-Emiratis, patients who had their visits in 2020, patients who had video visits, and those who sought family medicine as a specialty had a shorter turnaround time to receive their appointment. Conclusions: This study identifies several characteristics associated with the turnaround time. Moreover, technological improvements focusing on specialties that can readily be addressed through telehealth and further research in this domain will improve service provision and support building an evidence-base in the government sector of the emirate of Dubai.

## 1. Introduction

The use of Information and Communication Technologies (ICT) has helped improve access to healthcare services [1]. The effect of ICT on healthcare service delivery increased significantly during the novel coronavirus disease 2019 (COVID-19) pandemic [2]. The increasing use of telehealth services is a significant modality through which ICT influences healthcare service delivery. Telehealth refers to the use of electronic ICT to facilitate the delivery of healthcare between patients and healthcare providers over long distances [3]. Telehealth can be used to exchange information related to the diagnosis, treatment, and prevention of diseases and injuries between patients and healthcare professionals, or for research and educational purposes [4]. While telemedicine and telehealth are often used interchangeably, some studies have distinguished between the two terminologies; the former refers to clinical services provided by physicians, while the scope of the latter expands to include non-clinical services, such as tele-education for patients and care providers [5,6].

Many health systems worldwide quickly transitioned to telehealth services during the pandemic to sustain service provision and to protect patients and healthcare workers [7,8,9,10]. In the United States, for example, the share of telehealth visits in the total outpatient visits increased more than 13-fold—from less than 1% in 2019 to more than 13% in 2020 [11]. Similarly, many countries in the Gulf Cooperation Council region either adopted or activated telehealth services during the pandemic to contain the virus and to continue providing healthcare services [12,13]. Both the Kingdom of Saudi Arabia and Kuwait have substantially demonstrated the use of mobile applications instead of in-person visits and enhanced their telehealth systems’ infrastructure [14]. Though some countries in the region have assessed the utilization of telehealth services during the COVID-19 pandemic [12,13,14,15], studies related to the assessment of telehealth utilization within the United Arab Emirates (UAE) context have been limited.

Similarly, in the UAE, the pandemic has accelerated the application of telehealth and has become commonplace in health service provision in an outpatient setting [16,17]. The Dubai Health Authority (DHA), the government entity that oversees healthcare in the Emirate of Dubai, ensured continued healthcare service provision in its facilities during the pandemic [18]. In 2017, DHA issued its first clinical guidelines for the use of telehealth in Dubai health facilities, with revised standards and guidelines issued in 2021 [19]. It ensured that telehealth covered a wide range of services, including general health consultations, laboratory test requests, medication refills, COVID-19-related services, dental services, and follow-ups [20]. The ultimate goals were to maintain health system performance, to improve the quality of healthcare delivery, and to reduce the worsening of patients’ outcomes that may occur due to discontinuity of care [21,22].

While the use and adoption of telehealth in the UAE have increased significantly, information on the characteristics and community utilization of telehealth in the UAE, and particularly in the Emirate of Dubai, is limited [7,23,24]. In addition, to our knowledge there were no studies conducted at the Dubai level that examined the waiting time of the utilization of telehealth during the pandemic. We believe this study will contribute to building a base of evidence to expand the role of telehealth after the COVID-19 pandemic and to optimize its performance in the government sector. Therefore, this study aimed to examine the characteristics of telehealth visits and the sociodemographic characteristics of patients who utilized telehealth services in DHA facilities between 2020 and 2021. Moreover, the study aimed to examine patients who were more likely to complete their appointments, as well as the factors associated with the turnaround time (TAT) among telehealth services users at DHA facilities. We hypothesized that there is no difference in the waiting time among a patient’s socio-demographic and visit characteristics among the telehealth users in the Emirate of Dubai.

## 2. Materials and Methods

### 2.1. Study Setting, Design, and Data Source

This cross-sectional retrospective study uses the secondary data of patients who received care using the “Doctor for Every Citizen” (DEC) telehealth modality at the DHA between March 2020 and July 2021. The DEC, hereinafter referred to as telehealth services, enables all UAE citizens and residents to remotely receive a telehealth visit from an expert physician via a video/voice call, without the need to physically visit a health facility [19]. Moreover, DEC is accessible 24/7, aiming to achieve the highest levels of expert virtual health communication [19]. Although the service was launched in early 2020, it has been expedited by the COVID-19 pandemic [19]. The process of telehealth visit starts with an appointment booking, after which the patient logs in to their DHA application on their smartphone to select DEC. Patients are then automatically connected and prompted to start a video/voice call with the physician. Every patient who receives telehealth services at DHA has a record of the visit documented in their electronic medical records (EMRs). A total of 241,822 records were extracted retrospectively from March 2020 to July 2021, reviewed, analyzed, and included in this study. The DHA has a centralized EMR system that connects all government healthcare facilities including hospitals, primary healthcare, and specialized centers. All patient records in the EMR under the Doctor for Every Citizen initiative were extracted and analyzed for this study.

### 2.2. Variables and Measures

The EMRs comprised data related to patients’ socio-demographics and visit characteristics. The patients’ socio-demographic characteristic variables analyzed were age, gender, and nationality. Patients were grouped according to their age: under 18, 18–24, 25–44, 45–59, and 60 years and above. Nationality was dichotomized into Emirati (UAE citizens) and non-Emirati for all other nationalities. The visit characteristic variables analyzed included provider specialty, visit status, total number of visits, TAT, and appointment completion status. Visit types were grouped into audio-only (telephone) and audiovisual (video). Wait time or TAT was defined as the duration (in days) from the appointment request date to the date the patient received the telehealth service. Appointment completion status was categorized as canceled if the scheduled appointment did not turn into the actual care received (i.e., patient not seen by a physician) or completed otherwise (i.e., patient seen by a physician). Cancellation reasons were further grouped into patient-related reasons (e.g., financial reasons, positive COVID-19 diagnosis, or non-emergency complaints), provider-related reasons (e.g., physician attending conference, training, or meeting or on emergency leave), and technical system-related reasons (e.g., technology, resource maintenance, web/app cancellation, or system error). Providers’ specialties were listed by the frequency of visits for the five top consulted specialties, and the remaining specialty visits were grouped under ‘others’.

### 2.3. Statistical Analysis

Data coding, management, and analysis were conducted using IBM SPSS (Version 22.0, SPSS, IBM Corp, Armonk, NY, USA) and Stata (Version 17, Stata Corporation, College Station, TX, USA). As the data were extracted from EMR, no missing data were found, and the full data set was used in this analysis. Relative frequencies were reported for categorical variables. The chi-square test was used for binary and categorical variables to examine the association in a bivariate analysis. After testing for skewness and kurtosis, the median values for appointment TAT were calculated to minimize the effect of noise and outliers. The level of significance was set at 5% (*p* < 0.05), and confidence intervals (CIs) were calculated at 95%. A binary logistic regression model was constructed to examine the association between appointment TAT (≤2 days or ≥3 days) as an outcome of interest, with patients and visit characteristics as independent variables. Adjusted odds ratios (OR) were reported to reflect the strength of the association. The stepwise forward selection method was used to adjust for potential confounders in the model. The study protocol was approved by the Dubai Scientific Research Ethics Committee of the Dubai Health Authority (reference number DSREC-03/2022-08). 

## 3. Results

Table 1 shows the demographic characteristics of patients who scheduled telehealth visits according to their appointment status. Of the 241,822 scheduled telehealth visits, 189,951 (78.55%) were completed per schedule, and the remaining (21.45%) were incomplete or canceled. Among the patients who scheduled telehealth visits, 57.81% were females, 67.08% were Emirati, and 42.16% were aged 25–44 years (mean ± SD = 38.6 ± 19.5 years). The top reasons for using telehealth services at DHA during the COVID-19 pandemic are illustrated in Figure 1. Seeking providers’ consultation (11.8%) was the most common reason, followed by COVID-19 related (8.65%) and medication refills (4.65%).

Table 2 shows the visit characteristics of the patients who scheduled telehealth services according to the appointment status. Of the total scheduled appointments, 55.88% were scheduled in 2021. Regarding appointment TAT (wait time), 71.27% of the patients had to wait two days or less, and the remaining (28.73%) had to wait for three or more days from the booking date to see a provider. Of the total scheduled telehealth visits, 67.09% were audio–video visits. Family medicine was the most visited specialty (69.60%), followed by dental and oral surgery (9.23%), psychiatry and psychology (4.78%), and all other specialties (16.39%).

Additionally, the results from Table 1 and Table 2 highlight the differences between completed and canceled appointments for patients. Females (78.88%) completed more appointments than males (78.09%) did, and patients aged 18–24 years had the highest completed appointment rate (79.07) compared with other age groups. Emirati nationals (80.35%) completed their appointments more frequently than the non-Emiratis (74.87%) did. The appointments scheduled in 2020 (83.14%) had a higher completion frequency than those scheduled in 2021 (74.92%). Appointments with a shorter TAT (≤2 days) had higher completion rates (79.65%) than those with a TAT of ≥3 days (75.82%). Dental and oral surgery specialties had the highest appointment completion frequency (84.72%) among all other specialties. Figure 2 shows the reasons for cancelling the telehealth appointments. Patient-related and system-related reasons were reported in approximately half of incomplete cases. One out of ten (9.9%) cancellations was attributed to providers’ causes. The remaining patients (43.2%) were classified as undefined or unspecified.

Table 3 shows the demographic and visit characteristics of patients who completed telehealth services through TAT appointments. The median TAT was one to two days for many of the demographic and visit characteristics examined. However, some provider specialties, such as neurology, had the highest median TAT (26 days), followed by psychiatry and psychology (19 days). Similarly, males and females had fairly similar TAT values. Patients in the age group 25–44 years had a shorter TAT of ≤2 days compared with the other groups (77.85%). Non-Emiratis had a shorter TAT (≤2 days) than Emiratis (78.39%). Generally, TAT was shorter in 2020 (≤2 days) than in 2021 (88.65%). Patients who had video visits had a shorter TAT (≤2 days) than those who had telephone visits. Figure 3 shows the monthly trend and timeline of telehealth utilization during the COVID-19 pandemic. This shows that the trend of telehealth visits during the study period aligned well with the peak in COVID-19 cases in the UAE. In 2020, the highest number of telehealth visits was in April, whereas in 2021, most visits were scheduled in March and June.

Table 4 shows the results of the multiple logistic regression model with adjusted ORs to examine factors associated with TAT. The r-squared for the goodness-of-fit measure for our model was 40%. After adjusting for all the variables in the model, the oldest age group (≥60 years) had the lowest odds of waiting for more than two days to see a telehealth provider compared with the youngest age group of <18 years (OR 0.39, 95%CI:0.38, 0.42). Additionally, non-Emiratis had lower odds of waiting for more than two days to see a telehealth provider compared with Emiratis (OR 0.93, 95%CI:0.91, 0.96). In contrast, patients who used telehealth services during 2021 (OR 3.23, 95%CI:3.14, 3.32) and those who used the telephone modality (OR 7.28, 95%CI:7.05, 7.52) had higher odds of waiting for more than two days compared with those in 2020 and with those who had video visits, respectively. Regarding provider specialty, dental and oral surgery specialties had the highest odds of wait time (three or more days) compared with family medicine (OR 4.57, 95% CI 4.37, 4.79. All aforementioned predictors showed significant associations with appointment TAT at the level of *p* < 0.001.

## 4. Discussion

A higher proportion of patients who scheduled a telehealth appointment and completed their visits between 2020 and 2021 were females, Emiratis, and in the age group of 25–44 years. Additionally, more visits were completed in 2020, which had a shorter waiting time of two days or less from appointment requested to appointment received, were conducted via video, and sought family medicine as a specialty. Overall, older patients, non-Emiratis, patients who had their visits in 2020, patients who had video visits, and those who sought family medicine as a specialty had a shorter waiting time of two days or less to receive their appointment. Therefore, age, nationality, year of appointment date, type of visit, and provider specialty were sensitive factors to changes in TAT, from requested to received. Notably, consulting a healthcare provider was among the top reasons for using telehealth, with the highest peak of telehealth utilization being in March 2021.

Evidence demonstrates a positive attitude and general acceptance of telehealth services in the UAE population. Factors including socio-demographic and clinical characteristics were significantly associated with the utilization of telehealth services during the pandemic [25]. The socio-demographic characteristics of our findings are consistent with other findings conducted nationally and internationally, where telehealth users were more likely to be females and younger [25,26]. Another comparative study found that older patients may use telehealth more than visiting primary healthcare centers [15]. This can be explained by the fact that older people are encouraged to stay home and to use telehealth services for their safety. Moreover, our results illustrated that the top reasons for seeking telehealth services in the government sector were consultations, COVID-19-related, and medication refills, which is consistent with another study conducted in the UAE during the same period [25].

Some telehealth services could not be completed owing to many reasons. The reasons can be categorized as patient-related, technical system-related, provider-related, and other unknown reasons. Some studies attributed this to the unfamiliarity of patients with telehealth services, the need for physical examination, and limited insurance coverage. While insurance coverage is unlikely to be the case in our study, as the majority of our sample comprised UAE nationals who were covered by the government, the other two reasons need to be investigated [25,26]. Moreover, other reasons that could result in incomplete telehealth appointments include the healthcare model and how the services were provided, or the trust and confidence in the physician providing the service [25,26,27,28].

Regarding TAT, patients had to wait for a median of one day for a video visit. This seems to be a shorter waiting time compared with the time of an in-person visit before the COVID-19 pandemic in Dubai. One study illustrated that the average waiting time to see a physician at a primary healthcare center in Dubai was 35 days before the COVID-19 pandemic [29]. Moreover, our analysis confirms the findings from other studies, which revealed that patients seeing a family physician are less likely to wait compared with other specialties that follow the same pattern for in-person visits [30,31,32,33,34].

Although other specialties, such as dental and oral surgery, neurology, and dermatology, have longer waiting times, psychiatric and psychological services are crucial, especially in light of the mental health challenges and concerns associated with the pandemic. Psychiatry and psychology can be addressed through telemedicine, since less hands-on contact is needed with patients compared to other specialties [35]. However, other specialties may need more in-person medical attention. Therefore, ensuring that the psychological needs of a population are met by establishing adequate services and providers and mental health professionals per population is necessary. Research has demonstrated that mental health might need timely intervention since longer waiting times can have grave consequences on both the mental and physical health of patients and may worsen the symptoms of depression and lead to the development of self-harm ideation [25,26,36,37].

Our study has some limitations. The data were extracted only from the government sector in Dubai; therefore, they cannot be generalized to the private sector or other Emirates. Moreover, owing to the retrospective nature of the study, we were limited by the number of variables available for analysis. Notwithstanding these limitations, our study has several strengths, especially in that it has policy implications for the future use of telehealth services in the Emirate of Dubai. The study researchers grouped the factors into patients’ (socio-demographic characteristics) and healthcare system levels (visit characteristics) in relation to the TAT. Furthermore, this completion rate and TAT analysis have not been examined in a regional context before and have been minimally examined elsewhere. Additionally, the power of this study stems from the relatively large sample size and uniqueness of the dataset examined.

## 5. Conclusions

Our study, to the best of our knowledge, is among the few studies that examine the utilization of telehealth services, considering appointment completion and TAT in association with patients’ characteristics and visit characteristics in the Emirate of Dubai. Our results demonstrate that age group, nationality group, year of appointment date, visit type, and provider specialties are significant factors in understanding the utilization of telehealth services, especially during the COVID-19 pandemic. Overcoming the technological challenges is imperative when dealing with logistical hurdles experienced by end users. These improvements should consider evaluating users experience with the friendly use of telehealth services and the quality of information fed into the system. Enhance feeding of an information system will help extract and analyze better information, which will have significant policy implications. Physicians’ recommendations and best practices will help mitigate future technological challenges [38].

Telehealth has improved accessibility to health services. Numerous studies have discussed the cost-effectiveness of this technology in some specialties. Therefore, the government sector and health insurance companies may consider revising the cost and funding models related to telehealth services bearing in mind the different cost analysis types [39,40,41]. Given that our findings illustrate differences in turnaround time between specialties, this study supports the need for earlier tele-psychiatric assessment and interventions by healthcare professionals to contain mental health symptoms and potential suicidal thoughts among patients who require psychiatric and psychological services. Furthermore, cost-effectiveness, mixed-method designs, and qualitative studies are recommended to understand telemedicine utilization in the context of Dubai and the UAE.

## Figures and Tables

**Figure 1 healthcare-11-00294-f001:**
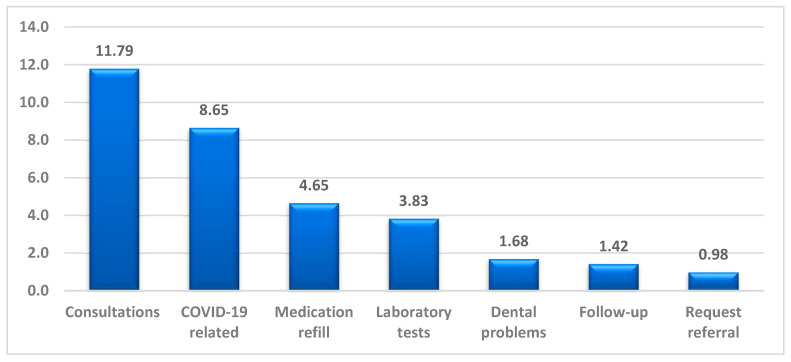
Top reasons for seeking telehealth visits at DHA from March 2020 to July 2021 (%).

**Figure 2 healthcare-11-00294-f002:**
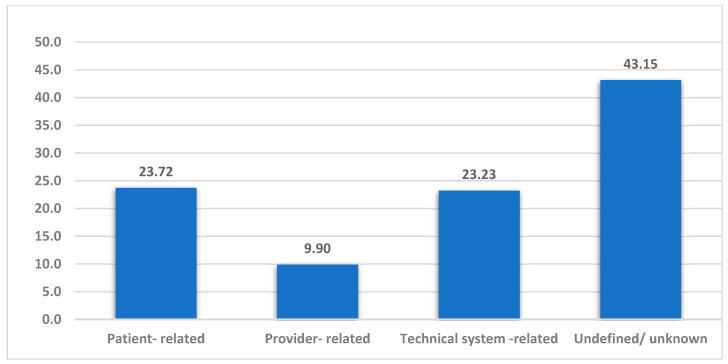
Reasons for canceled telehealth visits at DHA, March 2020–July 2021 (%).

**Figure 3 healthcare-11-00294-f003:**
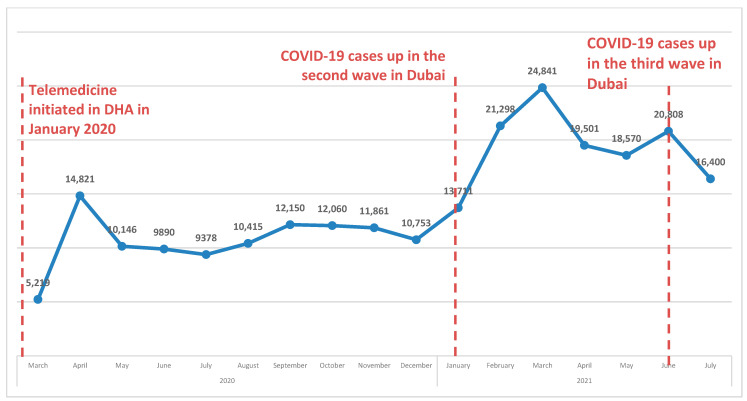
Trend/timeline of monthly telehealth utilization during the COVID-19 pandemic from March 2020 to July 2021.

**Table 1 healthcare-11-00294-t001:** Demographic characteristics of patients who scheduled a telehealth visit at DHA, by appointment status, March 2020–July 2021.

	Appointment Status			
	Completed N (Row %)	Canceled N (Row %)	Total N (Column %)	*p* Value *
Gender				
Male	79,667 (78.09)	22,351 (21.91)	102,018 (42.19)	<0.001
Female	110,284 (78.88)	29,520 (21.12)	139,804 (57.81)	
Age Group				
<18 years	27,440 (78.76)	7398 (21.24)	34,838 (14.41)	0.018
18–24 years	17,341 (79.07)	4590 (20.93)	21,931 (9.07)	
25–44 years	79,818 (78.28)	22,144 (21.72)	101,962 (42.16)	
45–59 years	35,429 (78.86)	9497 (21.14)	44,926 (18.58)	
≥60 years	29,923 (78.40)	8242 (21.60)	38,165 (15.78)	
Nationality Group				
Emirati	130,354 (80.35)	31,872 (19.65)	162,226 (67.08)	<0.001
Non-Emirati	59,597 (74.87)	19,999 (25.13)	79,596 (32.92)	
Total	189,951 (78.55)	51,871 (21.45)	241,822 (100.00)	

* *p* value for Chi square test significant at <0.05.

**Table 2 healthcare-11-00294-t002:** Visit characteristics of patients who scheduled a telehealth visit at DHA, by appointment status, March 2020–July 2021.

Appointment Status				
	Completed N (Row %)	Canceled N (Row %)	Total N (Column %)	*p* Value *
Appointment scheduled date (year)				
2020	88,708 (83.14)	17,985 (16.86)	106,693 (44.12)	<0.001
2021	101,243 (74.92)	33,886 (25.08)	135,129 (55.88)	
Appointment TAT				
≤2 days	137,281 (79.65)	35,076 (20.35)	172,357 (71.27)	< 0.001
≥3 days	52,670 (75.82)	16,795 (24.18)	69,465 (28.73)	
Visit type				
Video visits	127,382 (78.51)	34,862 (21.49)	162,244 (67.09)	0.523
Telephone visits	62,569 (78.63)	17,009 (21. 37)	79,578 (32.91)	
Provider specialty				
Family Medicine	132,230 (78.56)	36,078 (21.44)	168,308 (69.60)	<0.001
Dental & oral surgery **	18,918 (84.72)	3413(15.28)	22,331 (9.23)	
Psychiatry & Psychology	9387 (81.15)	2180 (18.85)	11,567 (4.78)	
Dermatology	6882 (73.38)	2496 (26.62)	9378 (3.88)	
Neurology	5685 (78.75)	1534 (21.25)	7219 (2.99)	
Others ***	16,849 (73.20)	6170 (26.80)	23,019 (9.52)	
Total	189,951 (78.55)	51,871 (21.45)	241,822 (100.00)	

* *p* value for Chi square test significant at <0.05; TAT; turnaround time; ** Dental and Oral Surgery includes Dental, Oral Maxillofacial Surgery, and Orthodontics; *** Others include the following specialties: Cardiology, Cardiothoracic Surgery, Endocrinology (Endocrinology and Diabetes), General Surgery (General Surgery, Bariatrics, Hand Surgery, and Plastic Surgery), Hematology (Hematology and Thalassemia), Home Health Services, Internal Medicine (Internal Medicine, Infectious Diseases, Geriatric Medicine, and Rheumatology), Nutrition (Dietitian and Nutrition), Neurosurgery, Obstetrics and Gynecology (Gynecology and Obstetrics), Oncology and Nuclear Medicine, Ophthalmology, Trauma, Orthopedic Surgery, Otolaryngology (Otolaryngology and Audiology), Pediatrics (Pediatric Gastroenterology and Pediatric Neurology), Physical Therapy and Rehabilitation (Occupational Therapy, Physical Therapy, and Rehabilitation), Pulmonology, and Vascular Surgery.

**Table 3 healthcare-11-00294-t003:** Patient and visit characteristics of patients who completed a telehealth visit at DHA, by turnaround time (TAT), March 2020–July 2021.

Appointment TAT in Days					
	Median TAT	≤2 Days N (Row %)	≥3 Days N (Row %)	Total N (Column %)	*p* Value *
Gender					
Male	1	57,485 (72.16)	22,182 (27.84)	79,667 (41.94)	0.034
Female	1	79,796 (72.36)	30,488 (27.64)	110,284 (58.06)	
Age Groups					
<18 years	2	17,335 (63.17)	10,105 (36.83)	27,440 (14.45)	<0.001
18–24 years	2	11,157 (64.34)	6184 (35.66)	1734 (19.13)	
25–44 years	1	62,138 (77.85)	17,680 (22.15)	79,818 (42.02)	
45–59 years	1	24,945 (70.41)	10,484 (29.59)	35,429 (18.65)	
60 + years	1	21,706 (72.54)	8217 (27.46)	29,923 (15.75)	
Nationality groups					
Emirati	1	90,565 (69.48)	39,789 (30.52)	130,354 (68.63)	<0.001
Non-Emirati	1	46,716 (78.39)	12,881 (21.61)	59,597 (31.37)	
Year of appointment date					
2020	1	78,633 (88.64)	10,075 (11.36)	88,708 (46.71)	<0.001
2021	2	58,648 (57.93)	42,595 (42.07)	101,243 (53.29)	
Visit type					
Telephone visit	10	17,585 (28.10)	44,984 (71.90)	62,569 (32.94)	<0.001
Video visit	1	11,9696 (93.97)	7686 (6.03)	127,382 (67.06)	
Provider’s specialty					
Family medicine	1	126,355 (95.55)	5875 (4.44)	132,230 (69.61)	<0.001
Dental and oral surgery **	8	3607 (19.06)	15,311 (80.93)	18,918 (9.95)	
Psychiatry and psychology	19	1598 (17.02)	7789 (82.97)	9387 (4.94)	
Dermatology	12	1224 (17.78)	5658 (82.21)	6882 (3.62)	
Neurology	26	706 (12.41)	4979 (87.58)	5685 (2.99)	
Others ***	8	3791 (22.49)	13,058 (77.51)	16,849 (8.87)	
Total	1	137,281 (72.27)	52,670 (27.23)	189,951 (100.00)	

* *p* value for Chi square test; * significant at <0.05 TAT; turnaround time; ** Dental and Oral Surgery includes Dental, Oral Maxillofacial Surgery, and Orthodontics; *** Cardiology, Cardiothoracic Surgery, Endocrinology (Endocrinology and Diabetes), General Surgery (General Surgery, Bariatrics, Hand Surgery, and Plastic Surgery), Hematology (Hematology and Thalassemia), Home Health Services, Internal Medicine (Internal Medicine, Infectious Diseases Geriatric Medicine, and Rheumatology), Nutrition (Dietitian and Nutrition), Neurosurgery, Obstetrics and Gynecology (Gynecology and Obstetrics), Oncology and Nuclear Medicine, Ophthalmology, Trauma, Orthopedic Surgery, Otolaryngology (Otolaryngology and Audiology), Pediatrics (Pediatric Gastroenterology and Pediatric Neurology), Physical Therapy and Rehabilitation (Occupational Therapy, Physical Therapy, and Rehabilitation), Pulmonology, and Vascular Surgery.

**Table 4 healthcare-11-00294-t004:** Binary logistic regression analysis for the factors associated with turnaround time for patients who completed telehealth visits at DHA, March 2020–July 2021.

Characteristics	Odds Ratio (OR)	(95% CI)	*p* Value
Gender			
Male	Reference		
Female	1.01	(0.98, 1.04)	0.513
Age Group			
<18 years	Reference		
18–24 years	0.83	(0.78, 0.87)	<0.001
25–44 years	0.70	(0.67, 0.73)	<0.001
45–59 years	0.64	(0.61, 0.67)	<0.001
≥60 years	0.39	(0.38, 0.42)	<0.001
Nationality groups			
Emirati	Reference		
Non-Emirati	0.93	(0.91, 0.96)	<0.001
Year of appointment date			
2020	Reference		
2021	3.23	(3.14, 3.32)	<0.001
Visit type			
Video visit	Reference		
Telephone visit	7.28	(7.05, 7.52)	<0.001
Provider specialty			
Family medicine	Reference		
Dental and oral surgery	4.57	(4.37, 4.79)	<0.001
Dermatology	3.73	(3.49, 3.99)	<0.001
Psychiatry and psychology	3.09	(2.92, 3.28)	<0.001
Neurology	3.81	(3.54, 4.10)	<0.001
Other specialties	3.26	(3.12, 3.41)	<0.001

CI, confidence interval at 95%; *p* value < 0.05.

## Data Availability

The data that support the findings of this study are from the Dubai Health Authority “SALAMA system”. However, restrictions apply to the availability of these data; thus, these data are not publicly available. Data are available from the corresponding author upon responsible request with permission from the Dubai Health Authority.

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
