# Peer review of "Doctor for Every Citizen: Telehealth Visits at Dubai Health Authority during COVID-19 Pandemic in Dubai, United Arab Emirates"

_healthcare, 2023, doi:10.3390/healthcare11030294_

Round 1
Reviewer 1 Report
Attached.

Author Response
PDF file is uploaded.

Reviewer 2 Report
In general, the paper has a good research design and is well-written. The comment below contains suggestions for improving the manuscript:
1. In the results section (paragraphs 188-200) there is redundant information that is repeated in table 4. It should be corrected so that the text contains only the main information about the analysis and that the same text refers to the table. Also in this part, it is necessary to specify additional parameters about the logistic regression model (whether the model is statistically significant and how much of the variance it explains)
2. It is necessary to technically correct the figures so that they have the same dimensions and alignment.
Author Response
PDF file is uploaded.

Reviewer 3 Report
The proposed study identified the factors that may determine the utilization of telehealth services in Dubai Health Authority (DHA) facilities and the factors associated with telehealth appointment completion and turnaround time.
Please respond to the following comments:
1. Authors need to clearly state the proposed study's problem statement, aim, hypotheses, and significance. For instance, a lack of information is insufficient as a problem statement. Authors need to construct a better argument for the problem statement. Similarly, please provide a rationale for selecting the input features for the model. I believe the following study will help to show the significance of the proposed study from the perspective of the impact of pandemic-related restrictions on mental health and how telehealth applications may tackle some of the challenges mentioned in the following study.
Mishra R, Park C, York MK, Kunik ME, Wung SF, Naik AD, Najafi B. Decrease in mobility during the COVID-19 pandemic and its association with increase in depression among older adults: A longitudinal remote mobility monitoring using a wearable sensor. Sensors. 2021 Apr 29;21(9):3090.
2. In the introduction, clarify the impact of missing appointments on the healthcare sector or “Doctor for Every Citizen Program”.
3. How was missing data handled? Were any imputation methods used? Or, if data were eliminated, provide the details about the included dataset.
4. Provide more clarity about the 241,822 records. Were they extracted from the single EMR or EMRs from the different hospitals or study sites combined? If EMRs were combined from different study sites, how did authors account for heterogeneity among EMRs? More details about EMR management under the Doctor for Every Citizen program are required.
5. What was the average duration of telehealth visits and Daily or monthly number of appointments fulfilled by the health care provider? This information will help to understand the burden on healthcare providers and the advantages of the telehealth program over in-person visits.
6. Authors mentioned, “Cancellation reasons were further grouped into patient-related reasons (e.g., financial reasons, positive COVID-19 diagnosis or non-emergency complaints), provider-related reasons (e.g., attending conference, training, meeting, or the physician went on emergency leave), and technical system-related reasons (e.g., technology, resource maintenance, web/app cancellation, or system error).” However, the authors should clearly state how the patient-related or physician-related reasons were used in the analysis. Were they used as the dependent or independent variable? This need to be mentioned in the method sections.
7. Authors should report the effect size in the results in tables 1 and 2.
Author Response
PDF file is uploaded.

Round 2
Reviewer 1 Report
It can be accepted with this version for me.